# Antimicrobial Susceptibility of *Staphylococcus aureus*, *Streptococcus agalactiae*, and *Escherichia coli* Isolated from Mastitic Dairy Cattle in Ukraine

**DOI:** 10.3390/antibiotics9080469

**Published:** 2020-08-01

**Authors:** Leta Elias, Ajay S. Balasubramanyam, Olena Y. Ayshpur, Iryna U. Mushtuk, Nataliya O. Sheremet, Volodymyr V. Gumeniuk, Jeffrey M. B. Musser, Artem S. Rogovskyy

**Affiliations:** 1Department of Veterinary Pathobiology, College of Veterinary Medicine and Biomedical Sciences, Texas A&M University, College Station, TX 77843, USA; le227@cornell.edu (L.E.); jmusser@cvm.tamu.edu (J.M.B.M.); 2College of Osteopathic Medicine, Kansas City University of Medicine and Biosciences, 2901 St. John’s Boulevard, Joplin, MO 64804, USA; abalasubramanyam@kcumb.edu; 3Institute of Veterinary Medicine of the National Academy of Agrarian Sciences of Ukraine, 30 Donetska Str., Kyiv 03151, Ukraine; olenaayshpur@gmail.com (O.Y.A.); mushtuk0104@gmail.com (I.U.M.); seremet-2@ukr.net (N.O.S.); Volodymyr.Gumeniuk@arterium.ua (V.V.G.)

**Keywords:** antimicrobial resistance, dairy farms, mastitis, bacterial isolates, Ukraine

## Abstract

Bovine mastitis is the predominant cause for antimicrobial use on dairy farms and is a major source of economic losses in the dairy industry. In this study, the antimicrobial susceptibility profiles of common mastitis-causing pathogens, *Staphylococcus aureus* (*n* = 62), *Streptococcus agalactiae* (*n* = 46), and *Escherichia coli* (*n* = 129), were determined for dairy cattle with mastitis across 142 Ukrainian farms. The results showed that there were more gentamicin resistant *S. aureus* isolates (16.95%) identified in this study than previously reported for Ukrainian dairy cattle. Moreover, low levels of amoxicillin susceptibly (13.51%) were observed for *St. agalactiae*, which contrasted a previous study showing susceptibility levels of >50%. *St. agalactiae* resistance to tetracycline was observed in 80% of the isolates. Cephalosporin use was most ineffective against *E. coli*, with 43.27–56% of the isolates exhibiting this resistant trait. Overall, this study performed a preliminary analysis of antimicrobial resistance on mastitis isolates from Ukrainian farms. However, given the limited numbers of the isolates tested in this study and that the publications on antimicrobial resistance in animal husbandry of Ukraine are very few, more extensive investigations are needed to comprehensively examine susceptibility patterns of mastitis-causing pathogens in dairy cattle in Ukraine.

## 1. Introduction

Bovine mastitis is one of the most prevalent conditions affecting dairy cattle and is the most frequent reason for antimicrobial use on dairy farms. Consequently, mastitis is a significant economic burden to the dairy industry [1,2]. Although *Escherichia coli*, *Staphylococcus aureus*, and *Streptococcus agalactiae* are considered some of the main mastitis-causing pathogens, many other bacterial species can be implicated [1,2,3,4]. Due to the bacterial diversity associated with bovine mastitis, and the fact that pathogen identification is not frequently performed for mastitic dairy cattle, antimicrobials with a broad spectrum of activity against both gram-negative and gram-positive organisms are routinely used in dairy medicine [1,5]. As a result of decades-long usage of antimicrobials to date, bacterial resistance is an increasing concern in, and beyond, veterinary medicine [2,6,7]. Monitoring susceptibility patterns of clinical isolates is a significant aspect of the One Health approach [8,9]. Some studies advocate the use of susceptibility testing for rationally selecting the most appropriate agents to counter mastitis infections, although the accuracy of in vitro susceptibility testing is limited in its ability to predict the curability of mastitis [1,10,11,12,13]. Nonetheless, it is recommended within the European Commission’s guidelines for the prudent use of antimicrobials in veterinary medicine that susceptibility testing be performed prior to mastitis treatment with antimicrobials to prevent the propagation of resistant bacteria via rationalized selection of an appropriate antimicrobial [10,11,14].

Given that there are almost no peer-reviewed publications on antimicrobial resistance in animal husbandry of Ukraine [15,16,17], the aim of this study was to assess the antimicrobial susceptibility patterns of mastitis-causing pathogens, specifically, *E. coli*, *S. aureus*, and *St. agalactiae* isolated from dairy cattle with mastitis in Ukraine.

## 2. Results

In this study, 13.0% (isolates/total isolates; 132/1017) of the isolates were identified as *S. aureus,* and susceptibility testing was performed for only 62 *S. aureus* isolates due to very limited resources. Gentamicin was mostly active against the *S. aureus* isolates, with a proportion of 16.95% (number of isolates/total isolates tested; 10/59) resistant isolates (Table 1). Against tetracycline, 21.43% (6/28) of the isolates were resistant. As for ofloxacin, 26.32% (5/19) of the *S. aureus* isolates were resistant. Antimicrobial susceptibility testing was also performed for the cephalosporins ceftiofur and cefotaxime. Against these cephalosporins, 41.51% (22/53) and 47.06% (8/17) of the isolates showed resistance to ceftiofur and cefotaxime, respectively (Table 1).

*St. agalactiae* accounted for 11.0% (112/1017) of the total isolates collected in this study, and of those, 46 were tested for antimicrobial susceptibility. Susceptibility testing revealed that a low proportion (13.51%; 5/37) were susceptible to amoxicillin (Table 1). Cefotaxime was moderately effective against *St. agalactiae*, as susceptibility was observed in 41.18% (7/17) of the isolates tested. Against ceftiofur, the proportion of resistant isolates was 25.58% (11/43). Ofloxacin was mostly effective against *St. agalactiae*, with a moderately low proportion of the isolates being resistant to this antimicrobial (17.65%; 3/17) (Table 1). Lastly, there were no Clinical and Laboratory Standards Institute (CLSI) interpretive criteria for gentamicin, but the zone diameters were included in Appendix A.

*E. coli* was identified in 23.12% (235/1017) of the total bacterial isolates collected in this study, and susceptibility testing was performed for 129 isolates (Table 1). Amoxicillin had poor activity against the *E. coli* isolates, with 77.45% (79/102) testing resistant to this antimicrobial. Cefotaxime resistance was present in over half of the isolates tested (56.25%; 27/48). In comparison, ceftiofur was slightly more active against *E. coli*, with 43.27% (45/104) being resistant to this drug (Table 1). Of the isolates tested against ofloxacin, 26.92% (14/52) exhibited resistance. Tetracycline was mostly effective against *E. coli*, with 18.75% (9/48) of the isolates showing this resistant trait. Against gentamicin, 26.27% (31/118) of the isolates were resistant to this antimicrobial (Table 1).

## 3. Discussion

Our results indicated that there was a wide variability in the antimicrobial susceptibilities of *S. aureus*, *St. agalactiae*, and *E. coli* isolated from dairy cattle with mastitis in Ukraine. Overall, the three pathogens exhibited a high level of resistance to the beta-lactam antimicrobials, amoxicillin, cefotaxime, and ceftiofur. In this study, only 13.51% of the *St. agalactiae* isolates were susceptible to amoxicillin. In comparison, the previous Ukrainian bovine mastitis study demonstrated that approximately half of the *St. agalactiae* isolates were susceptible to amoxicillin [15].

The resistance of the mastitis isolates to amoxicillin and the cephalosporins, cefotaxime and ceftiofur, was high for *S. aureus* and *E. coli*. This was comparable to reports for *S. aureus* from mastitis cases recorded for Europe and the United States, where beta-lactamase producers were identified in 35.1% of *S. aureus* isolates [18]. In contrast, a previous mastitis antimicrobial resistance study performed across European dairy farms reported that *S. aureus* and *E. coli* resistance to ceftiofur was identified in only 1% of the isolates [19]. To date, despite a growing prevalence of resistant bacteria, amoxicillin continues to be among the four most commonly used antimicrobials in veterinary medicine to treat *E. coli* and *S. aureus* infections in Ukraine [20].

The high proportion of ceftiofur- and cefotaxime-resistant *E. coli* isolates identified in the current study suggested that extended-spectrum beta-lactamase producing strains may be circulating across the investigated dairy farms, an important knowledge gap that warrants further investigation. This finding could potentially be explained by the unrestricted use of extended-spectrum cephalosporins in rural farming of Ukraine, and more specifically by the preferred use of these antimicrobials for treatment of bovine mastitis [21,22].

In light of the significant health threat posed by methicillin-resistant *S. aureus* (MRSA) in Europe, as well as globally, monitoring the prevalence of *S. aureus* infections among production animals would be warranted [23,24]. To date, only three cases of MRSA have been reported by a previous dairy cattle study within Dnipropetrovsk and Donetsk regions, Ukraine [17]. Unfortunately, there is no program to monitor the prevalence of staphylococcal infections among animals in Ukraine to date.

Gentamicin is among the leading antimicrobials used to combat animal infections, in general, of *S. aureus* and *E. coli* in Ukraine [20]. The overall effectiveness of gentamicin, as demonstrated in this study by the in vitro susceptibility of the mastitis isolates, may be a reason for its usage. Gentamicin usage in food animals, however, is carefully regulated in the European Union, and the European Commission has set limits on the allowable gentamicin residues found in animal food products [25]. It is worth noting that whereas 16.95% of the *S. aureus* isolates were resistant to gentamicin in this study, much fewer resistant *S. aureus* isolates were identified by a previous Ukrainian study in which only 2 of 62 mastitis isolates tested resistant to this aminoglycoside [16]. In agreement with this earlier Ukrainian bovine mastitis study, in the current investigation, tetracycline effectively suppressed *S. aureus* growth in vitro [16]. In the present study, tetracycline was also found to be highly efficacious against *E. coli* isolates, whereas resistance to this antimicrobial was frequently observed among *St. agalactiae*, with 80% of the isolates demonstrating this resistance trait. The latter is in concordance with previous studies, where resistance to tetracycline was frequently detected among *St. agalactiae* isolates [15,26,27]. Although the antimicrobial susceptibility profile of *St. agalactiae* was difficult to evaluate due to a lack of CLSI breakpoints, in general it is considered highly responsive to almost all antimicrobial treatment regimens [1].

Unfortunately, in Ukraine to date, there are no established mastitis treatment protocols, despite the fact that attention in this area has indeed grown since the country joined the World Trade Organization (WTO) in 2008 [28]. Additionally, prior to 2018, antimicrobial usage in Ukraine was mostly unregulated, as there was no national body or organization overseeing the consumption and application of these agents [29]. Given that there is a very limited number of studies that have evaluated the antimicrobial susceptibility patterns of *S. aureus*, *St. agalactiae*, and *E. coli* isolated from mastitic dairy cattle from Ukrainian farms, future investigations are highly warranted to assess and monitor in real time the status of antimicrobial resistance in animal husbandry of Ukraine, and to observe the effect of the recent antimicrobial policy initiatives in the future.

## 4. Materials and Methods

In the present study, bacteria were cultured from milk samples of mastitic dairy cattle as part of a routine diagnostic service provided to dairy farms in Ukraine. Between 2016 and 2018, a total of 142 farms located across various regions of Ukraine were included in this study. These year-round farms had a herd size of 500–4500 cattle (Holstein Friesians, Red Steppe, Ayrshire, Simmental, Black Red, and Jersey), of which 200–2000 animals were milking cows. The operations were equipped with various types of parlors and other milking equipment (Bratslav, Ukraine, DeLaval, Sweden; GEA, Germany).

Primary isolation was performed by inoculation of 10 μL of milk to Trypticase soy agar with 5% bovine blood and 0.1% esculin, and the cultures were incubated for 48 h at 37 °C. The genus and species of the isolates were identified based on colony morphology, Gram stain, and biochemical analysis (API^®^ 20E, BioMeriex, France). In total, 1017 bacterial isolates were identified, including *E. coli* (*n* = 235), *S. aureus* (*n* = 132), and *St. agalactiae* (*n* = 112). Unfortunately, due to limited resources, only 129 *E. coli* isolates from 88 farms, 68 *S. aureus* isolates from 55 farms, and 46 *St. agalactiae* isolates from 44 farms were tested for antimicrobial susceptibility. Antimicrobial susceptibility testing was performed on randomly selected isolates via the disc diffusion method [30] for the following antimicrobial agents: amoxicillin (20 μg), cefotaxime (30 μg), ceftiofur (30 μg), gentamicin (10 μg), ofloxacin (5 μg), and tetracycline (30 μg) (Pharmactiv, Ukraine). The test results were interpreted by using human breakpoints for all the antimicrobial agents except for ceftiofur, for which veterinary interpretive criteria for cattle were used (Appendix A) [31,32].

## Figures and Tables

**Table 1 antibiotics-09-00469-t001:** Antimicrobial susceptibility of bacterial isolates from Ukrainian dairy cattle with mastitis.

*Staphylococcus aureus* Isolates							
Antimicrobials *	Isolates tested	# Resistant	% Resistant	# Susceptible	% Susceptible	# Intermediate	% Intermediate
Amoxicillin	57	–	–	–	–	–	–
Cefotaxime **	17	8	47.06	4	23.53	5	29.41
Ceftiofur	53	22	41.51	17	32.08	14	26.42
Gentamicin	59	10	16.95	46	77.97	3	5.08
Ofloxacin	19	5	26.32	11	57.89	3	15.79
Tetracycline	28	6	21.43	14	50.00	8	28.57
***Streptococcus agalactiae* Isolates**							
Antimicrobials *	Isolates tested	# Resistant	% Resistant	# Susceptible	% Susceptible	# Intermediate	% Intermediate
Amoxicillin	37	–	–	5	13.51	–	–
Cefotaxime	17	–	–	7	41.18	–	–
Ceftiofur	43	11	25.58	20	46.51	12	27.91
Gentamicin	39	–	–	–	–	–	–
Ofloxacin	17	3	17.65	8	47.06	6	35.29
Tetracycline	10	8	80.00	2	20.00	0	0.00
***Escherichia coli* Isolates**							
Antimicrobials *	Isolates tested	# Resistant	% Resistant	# Susceptible	% Susceptible	# Intermediate	% Intermediate
Amoxicillin	102	79	77.45	9	8.82	14	13.73
Cefotaxime	48	27	56.25	6	12.50	15	31.25
Ceftiofur	104	45	43.27	22	21.15	37	35.58
Gentamicin	118	31	26.27	56	47.46	31	26.27
Ofloxacin	52	14	26.92	34	65.38	4	7.69
Tetracycline	48	9	18.75	32	66.67	7	14.58

(*) The interpretation of minimum inhibitory zones was based on the 2019 Clinical and Laboratory Standards Institute (CLSI) guideline M100 with the exception of ceftiofur, for which veterinary CLSI breakpoints (CLSI guideline VET08) were used. (**) Susceptibility or resistance of *S. aureus* isolates to cefotaxime was predicted by interpreting the respective results for ceftiofur. (–) Interpretation was not performed because there were no clinical breakpoints.

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
