# Peer review of "Antimicrobial Susceptibility of Staphylococcus aureus, Streptococcus agalactiae, and Escherichia coli Isolated from Mastitic Dairy Cattle in Ukraine"

_antibiotics, 2020, doi:10.3390/antibiotics9080469_

Round 1
Reviewer 1 Report
Thank you for the opportunity to review the manuscript Antimicrobial susceptibility of Staphylococcus aureus, Streptococcus agalactiae, and Escherichia coli isolated from mastitic dairy cattle in Ukraine.
Overall this manuscript is very well written. Specific minor revisions are recommended.
Introduction
Line 50: Please cite the known literature here for Urkrainian specific antimicrobial use publications.
Results
- Please include the number of isolates that were multi-antimicrobial resistant (MAR) and multi-drug resistant (MDR). Please also include the MAR and MDR index descriptive statistics where applicable.
- Please include the breakpoint used in the text. As disc diffusion was used MIC data cannot be derived, so providing the breakpoints used is necessary to provide confidence in the designation.
- Please provide descriptive data for the number of farms with AMR and the number of isolates per farm. I acknowledge that the number of isolates compared to the number with susceptibility data is vastly different, however, some evidence (or lack) for farm effect is important.
- If available somatic cell count data should also be included.
Methods
- Please include any descriptive data about the size of herd, breeds present, production system type (such as seasonal, year round production etc), milking equipment used, milking protocols.
Discussion
Line 121-126- Please include a statement about the use of this medication in other jurisdictions. This section would lead the reader to believe that gentamicin is an ideal choice. While in the strict sense this is true, it is prohibited for use in food producing animals in many areas.
Author Response
Reviewer 1:
Overall this manuscript is very well written. Specific minor revisions are recommended.
Introduction
Line 50: Please cite the known literature here for Urkrainian specific antimicrobial use publications.
The citations were included per reviewer’s recommendation (see page 2 line 51).
Results
Please include the number of isolates that were multi-antimicrobial resistant (MAR) and multi-drug resistant (MDR). Please also include the MAR and MDR index descriptive statistics where applicable.
The authors would like to refrain from indicating the percentage of MDR/MAR or MDR/MAR indexing as numbers of antimicrobial classes and the numbers of isolates used for this preliminary study were very limited and quite variable.
Please include the breakpoint used in the text. As disc diffusion was used MIC data cannot be derived, so providing the breakpoints used is necessary to provide confidence in the designation.
The breakpoints used in this study are now provided in Table S2, which is now mentioned on page 4 line 189.
Please provide descriptive data for the number of farms with AMR and the number of isolates per farm. I acknowledge that the number of isolates compared to the number with susceptibility data is vastly different, however, some evidence (or lack) for farm effect is important.
The number of isolates and the number of farms are now indicated in the Material and Methods results (see page 4 lines 173-177).
If available somatic cell count data should also be included.
Unfortunately, this information was not available.
Methods
Please include any descriptive data about the size of herd, breeds present, production system type (such as seasonal, year round production etc), milking equipment used, milking protocols.
Additional information on the farms that participated in this study is now included in the materials and methods section (see page 4 lines 173-177).
Discussion
Line 121-126- Please include a statement about the use of this medication in other jurisdictions. This section would lead the reader to believe that gentamicin is an ideal choice. While in the strict sense this is true, it is prohibited for use in food producing animals in many areas.
The respective statement is now included (see page 4 lines 147-149).
Reviewer 2 Report
Comments to “Antimicrobial susceptibility of Staphylococcus aureus, Streptococcus agalactiae, and Escherichia coli isolated from mastitic dairy cattle in Ukraine ”. Bovine mastitis is the predominant cause for antimicrobial use on dairy farms and is a major source of economic losses in the dairy industry. The authors evaluated the antimicrobial susceptibility profiles of common mastitis-causing pathogens from Ukrainian farms. Overall, this study performed a preliminary analysis of antimicrobial resistance on mastitis isolates from Ukrainian farms. The research is useful and important for the bovine mastitis because this investigation usually is few in different countries. It is suggested that the article might be revised before it can be published. The following questions are the specific ones:
1.Line 131, “...frequently detected among S. agalactiae isolates (20%; 8/10)”. It is not clear of the meaning of “8/10”.
2.The authors used the CLSI breakpoint to tell the resistance. My suggestion is that the authors should offered the concentration of the tested drugs, and the raw data for drug resistance, which might be useful for other researchers to compare the drug resistance.
Author Response
Comments to “Antimicrobial susceptibility of Staphylococcus aureus, Streptococcus agalactiae, and Escherichia coli isolated from mastitic dairy cattle in Ukraine ”. Bovine mastitis is the predominant cause for antimicrobial use on dairy farms and is a major source of economic losses in the dairy industry. The authors evaluated the antimicrobial susceptibility profiles of common mastitis-causing pathogens from Ukrainian farms. Overall, this study performed a preliminary analysis of antimicrobial resistance on mastitis isolates from Ukrainian farms. The research is useful and important for the bovine mastitis because this investigation usually is few in different countries. It is suggested that the article might be revised before it can be published. The following questions are the specific ones:
1.Line 131, “...frequently detected among S. agalactiae isolates (20%; 8/10)”. It is not clear of the meaning of “8/10”.
The (20%; 8/10) was in reference to number and percent of tetracycline resistant St. agalactiae isolates identified in this study. This redundancy was removed from the text (page 4 line 157).
2.The authors used the CLSI breakpoint to tell the resistance. My suggestion is that the authors should offered the concentration of the tested drugs, and the raw data for drug resistance, which might be useful for other researchers to compare the drug resistance.
The concentrations of the discs used in this study are now provided on (see page 4 lines 185-187). The raw data are presented in Table S1.